# Domain Adaptation using Silver Standard Labels for Ki-67 Scoring in Digital Pathology: A Step Closer to Widescale Deployment

Amanda Dy[*1]                                          AMANDA.DY@TORONTOMU.CA
Ngoc-Nhu Jennifer Nguyen[2]               NGOCNHUJENNIFER.NGUYEN@MAIL.UTORONTO.CA
Seyed Hossein Mirjahanmardir[1]                    SHMIRJAHANMARDI@RYERSON.CA
Melanie Dawe[3]                                    MELANIE.DAWE@UHNRESEARCH.CA
Anthony Fyles[3]                                      ANTHONY.FYLES@RMP.UHN.CA
Wei Shi[3]                                               W.SHI@UHNRESEARCH.CA
Fei-Fei Liu[3]                                          FEI-FEI.LIU@RMP.UHN.CA
Dimitrios Androutsos[1]                                   DIMITRI@TORONTOMU.CA
Susan Done[3]                                           SUSAN.DONE@UHN.CA
April Khademi[1,4,5]                                   AKHADEMI@TORONTOMU.CA

[1] *Electrical, Computer, & Biomedical Engineering, Toronto Metropolitan University, Toronto, CAN*
[2] *Department of Laboratory Medicine and Pathobiology, University of Toronto, Toronto, ON, CAN*
[3] *Princess Margaret Cancer Centre, University Health Network, Toronto, ON, CAN*
[4] *Keenan Research Center, St. Michael's Hospital, Toronto, ON, CAN*
[5] *Institute of Biomedical Engineering, Science and Technology (iBEST), Toronto, ON, CAN*

**Editors:** Accepted for publication at MIDL 2023

## Abstract

Deep learning systems have been proposed to improve the objectivity and efficiency of Ki-67 PI scoring. The challenge is that while very accurate, deep learning techniques suffer from reduced performance when applied to out-of-domain data. This is a critical challenge for clinical translation, as models are typically trained using data available to the vendor, which is not from the target domain. To address this challenge, this study proposes a domain adaptation pipeline that employs an unsupervised framework to generate silver standard (pseudo) labels in the target domain, which is used to augment the gold standard (GS) source domain data. Five training regimes were tested on two validated Ki-67 scoring architectures (UV-Net and piNET), (1) SS Only: trained on target silver standard (SS) labels, (2) GS Only: trained on source GS labels, (3) Mixed: trained on target SS and source GS labels, (4) GS+SS: trained on source GS labels and fine-tuned on target SS labels, and our proposed method (5) SS+GS: trained on source SS labels and fine-tuned on source GS labels. The SS+GS method yielded significantly ($p < 0.05$) higher PI accuracy (95.9%) and more consistent results compared to the GS Only model on target data. Analysis of t-SNE plots showed features learned by the SS+GS models are more aligned for source and target data, resulting in improved generalization. The proposed pipeline provides an efficient method for learning the target distribution without manual annotations, which are time-consuming and costly to generate for medical images. This framework can be applied to any target site as a per-laboratory calibration method, for widescale deployment.

**Keywords:** Ki-67, proliferation index, domain adaptation, self-supervised learning

## 1. Introduction

Breast cancer is the most diagnosed cancer and the leading cause of cancer-related death in women worldwide (Sung et al., 2022). Ki-67 immunohistochemistry (IHC) biomarker is gaining traction for evaluating the proliferation rate of invasive breast cancers (Davey et al., 2021; Dowsett et al., 2011). Ki-67 expression is related to prognosis and can identify high-risk early-stage breast cancers (Penault-Llorca and Radosevic-Robin, 2017; Dowsett et al., 2011) and determine treatment modalities (Penault-Llorca and Radosevic-Robin, 2017; Davey et al., 2021; Dowsett et al., 2011). The Ki-67 proliferation index (PI) is the score associated with the proportion of Ki-67$^+$ tumour cells to the total number of tumour cells in a breast tissue section (Walters et al., 2013). However, quantifying this biomarker is labour-intensive, time-consuming, and subject to poor visual estimation concordance (Walters et al., 2013; Davey et al., 2021).

Fortunately, Ki-67 PI can be calculated with deep learning nuclei detection algorithms for more efficient and objective quantification. There have been a few deep learning tools addressing automated Ki-67 PI scoring in literature, such as piNET (Geread et al., 2021) and UV-Net (Mirjahanmardi et al., 2022) which were specifically developed for Ki-67 PI quantification in breast cancer. As automated artificial intelligence (AI) tools become more robust, there is a chance for translation and deployment. However, a challenge with widescale adoption is performance degradation at deployed target sites resulting when the target data is from a center that is not included in the (source) training set. It is especially evident in digital pathology given the variation in patient factors, specimen processing, staining protocols and acquisition devices across pathology laboratories. Annotations from target sites could be included in training sets, but generating gold standard (GS) ground truths are laborious and expensive for medical imaging.

Mitigating domain shift has become a topic of extensive research (Wang and Deng, 2018; Guan and Liu, 2022; Zou et al., 2020) and unsupervised domain adaptation (UDA) is gaining considerable attention for this task. UDA methods seek to overcome the domain gap without the need for labelled target data. Self-training (pseudo label-based methods) has emerged as a promising UDA solution (Natarajan et al., 2013; Zou et al., 2018). Self-training generates a set of pseudo labels in the target domain and re-trains a network based on these pseudo labels. Self-training loss encourages cross-domain feature alignment by learning from the labelled source data and pseudo-labelled target data. Pseudo labels can be quickly generated for any number of datasets, which is cost-effective and reduces development time. However, perfect accuracy cannot be guaranteed which can lead to propagated errors when fine-tuning. Because pseudo labels do not capture detailed features as well as clean labels we hypothesize that pre-training a network on pseudo labels from the target domain will allow a network to first learn dataset-specific characteristics and low-level features that are task-dependent, thereby providing optimal parameter initialization. Fine-tuning with GS (clean) labels from the source domain can then allow more detailed features to be captured by the network. This work proposes a pipeline that (1) uses an unsupervised Ki-67 PI quantification algorithm to generate pseudo labels, which we call silver standard (SS) labels, in the unlabeled target domain, (2) pre-trains a network on SS labels, and (3) fine-tunes the network on GS labels from the source domain. This pipeline can be used to calibrate automated deep learning-based medical imaging tools on a per-dataset basis, in an easy and

unsupervised manner. We validate our method on 325 clinical tissue microarrays (TMAs) (20800 patches) from the target domain. Experimental results show the proposed approach achieves superior performance on the pixel-level and patient-level, therefore, providing a DA training method for robust and accurate Ki-67 PI estimation.

## 2. Methods

### 2.1. Deep Learning Models

Two deep learning architectures are used for experiments: UV-Net and piNET, both developed for Ki-67 PI quantification in breast cancer and validated on large multi-institutional datasets. piNET was built using the U-NET architecture with an extra layer (Geread et al., 2021) and UV-Net was designed to preserve nuclear features of clustering or overlapping nuclei through dense 'V' blocks to retain the high-resolution details (Mirjahanmardi et al., 2022). The output of piNET and UV-Net is a multi-channel probability map, with center locations of tumour nuclei detected for two classes: Ki-67$^-$ and Ki-67$^+$ cells.

### 2.2. Transfer Learning

Transfer learning (TL) (Pan and Yang, 2010; Weiss et al., 2016) has proven to be effective for many real-world applications by exploiting knowledge in labelled training data from a source domain. TL has made major contributions to medical image analysis as it overcomes the data scarcity problem as well as preserving time and hardware resources. In this study, we introduce a TL approach that uses an unsupervised Ki-67 nuclei detection scheme to generate SS labels in the target domain for pre-training the model. This enables the model to learn the low-level nuclei features and attain optimal parameter initialization. We will then fine-tune the model using gold GS labels to capture more precise details and improve the accuracy of the learned features. We compare the performance of two network architectures, UV-Net and piNET, in the following scenarios: (1) pre-training with GS labels and fine-tuning with SS labels, and (2) pre-training with SS labels and fine-tuning with GS labels. The results are compared against training methods without TL.

### 2.3. Pseudo Label Generation: Silver Standards

In UDA settings, there are no labels for the target domain. Our goal is to improve performance on the target, so we train the model with the target SS labels generated by a previously developed and validated unsupervised Ki-67 nuclei detection method called the immunohistochemical colour histogram (IHCCH). The process includes vector median filtering, background subtraction, an unsupervised colour separation method that separates blue and brown objects automatically based on the histogram of the b* channel, and adaptive radius nuclei detection. More details can be found in (Geread et al., 2019).

### 2.4. Dataset

This study uses Ki-67 stained invasive breast cancer images obtained from three institutions. Table 2 summarizes the Ki-67 datasets used for each training method.

**Source Dataset:** 510 patches of 256×256 pixels in size are extracted from whole slide images provided by St. Michael's Hospital (SMH) in Toronto and an open-source database, Deepslide (Senaras, 2018). The ×20 Aperio AT Turbo and ×40 Aperio ScanScope scanners were used, respectively. Deepslide images are down-sampled to ×20 for compatibility. Images were annotated by marking Ki-67$^-$ and Ki-67$^+$ centroids (Geread et al., 2021). Centroid annotations were recast into a Gaussian kernel to allow the system to contextual learn information from the nuclei to help the classifier discover more robust features. Artifacts including overstaining, background, folders, blur, and dust are common in tissue slides; therefore, 15% of the training dataset includes patches with artifacts and non-tumorous areas to reduce false positives. This dataset represents our source domain and contains GS labels. Each patch contains 58 tumourous cells on average for a total of 29571 cells.

**Target Dataset:** The target dataset was provided by the University Health Network (UHN) and contains 411 tissue microarrays (TMA) from 175 patients. Each patient has 1 to 3 corresponding TMAs of 2000 × 2000 pixels in size and an expert PI estimate is available for each patient. 24 TMAs from 24 patients were used to create the SS labels. These 24 TMAs were tiled into patches of size 256x256 pixels and 345 patches which contained $\geq 80\%$ tumorous tissue were extracted and the remaining patches were discarded. The TMAs from patients used for SS label generation were removed from our target dataset to prevent patient data leakage. 10 TMAs were randomly selected from the remaining pool and annotated by an anatomical pathology resident (N.N.J.N) and verified by a breast pathologist (S.D.) to produce pixel-wise nuclei annotations for testing in the target domain. Each annotated TMA contains 2093 tumourous cells on average for a total of 20930 cells. Accordingly, the target domain test set contains 325 TMAs from 151 patients with patient-level PI scores and 10 TMAs with nuclei annotations.

### 2.5. Evaluation Metrics

Nuclei detection is evaluated by comparing the Ki-67$^-$ and Ki-67$^+$ centroids between the AI prediction and GS ground truths through the F1 score. The F1 score is the harmonic mean of precision and recall which is dependent on the number of true positives (TP), false positives (FP), and false negatives (FN). A TP is detected whenever the Euclidean distance between an annotation centroid and a detected centroid is less than 6 µm. This value corresponds to the average radius of tumourous cells from the source dataset. All detected cells not within 6 µm of a ground truth annotation are considered FP. Multiple detections of an already counted cell are also counted as FP. All ground truth cells without a detection within 6 µm proximity are considered FN. The F1 scores report raw nuclei detection performance, therefore, if a model is operating on an image with a low tumour nuclei count, a single missed nucleus can greatly skew the overall F1 score. Thus, different metrics, such as the proliferation index (PI) error should also be used. Tumour proliferation is measured by:

$$PI = \frac{\# \ Ki-67^+ \ tumour \ cells}{\#(Ki-67^+ \ + \ Ki-67^-) \ tumour \ cells} \tag{1}$$

which is computed over the whole TMA based on the detected nuclei. The PI difference is used to investigate the error between predicted and actual PI values: $\Delta PI = |PI_{actual} - PI_{predicted}|$. Pairwise one-way ANOVA is used to compare model performance.

### 2.6. Experimental Setup

Five training methods are used to study Ki-67 nuclei detection and PI estimation accuracy. The first configuration is GS only, which uses only the GS data from the source domain. The second configuration, SS only, uses the SS data generated by the unsupervised IHCCH algorithm from the target domain. The third configuration, Mixed, includes both GS and SS in the training pool. The fourth configuration, GS+SS, uses GS for pre-training and SS for fine-tuning and the final configuration is our proposed method, SS+GS, which uses SS for pre-training and GS for fine-tuning. All methods that use SS are trained with increments of 100 where each increment contains SS from previous increments. Table 2 summarizes the configurations of each training method. The IHCCH (unsupervised) method is also evaluated to verify the stand-alone performance of the tool. To ensure robustness to training variations we use a 3-fold cross-validation protocol for all experiments. We divide our 510 source patches with GS annotations into 3 subsets. For each fold, we select one subset as the held-out patches and the other 340 patches are used in the training pool. An Adam optimizer was used with a learning rate of 1e-3, a batch size of 4 with 100 epochs, and a Huber loss function, the epoch with the lowest validation loss was saved. Data augmentations were applied for rotation and scaling. All experiments were run using a GeForce RTX 3070 Ti.

## 3. Results

Quantitative results are summarized in Table 1. Nuclei predictions are shown in Figure 6. Reproducibility (standard deviation between 3-fold cross-validation models when predicting on the same target distribution data) is shown in Table 3.

Table 1: Average performance on held-out source and unseen target data over 3-fold cross-validation and all SS increments. Asterisk denotes significantly higher ($p < 0.05$) performance than baseline (GS Only) and bold denotes the best performance.

| Method | UV-Net | | | piNET | | |
| | F1 Source | F1 Target | $\Delta$PI | F1 Source | F1 Target | $\Delta$PI |
| --- | --- | --- | --- | --- | --- | --- |
| GS | 0.65±0.12 | 0.62±0.10 | 7.5±5.7 | 0.64±0.12 | 0.65±0.06 | 7.8±6.1 |
| IHCCH | 0.53±0.14 | 0.57±0.09 | 5.7±5.7* | 0.53±0.14 | 0.57±0.09 | 5.7±5.7* |
| SS | 0.55±0.11 | 0.62±0.09 | 6.9±6.1 | 0.57±0.10 | 0.68±0.08 | 6.8±6.1 |
| Mix | 0.64±0.13 | 0.71±0.07* | 6.0±5.6 | 0.64±0.11 | 0.71±0.08 | 6.1±5.7 |
| GS+SS | 0.56±0.10 | 0.62±0.07 | 6.6±6.0 | 0.57±0.11 | 0.65±0.07 | 6.3±5.7 |
| SS+GS | **0.67±0.12** | **0.76±0.08*** | **4.1±3.7*** | **0.65±0.12** | **0.77±0.08*** | **4.8±4.2*** |

### 3.1. Source Domain: Nuclei Detection

170 unseen patches from the source domain with pixel-level Ki-67$^-$ and Ki-67$^+$ centroid annotations are used to test nuclei detection performance. The distributions of the F1 scores are shown in Figure 4 and summarized in Table 1. The proposed SS+GS method yielded superior or competitive F1 performance on the source domain when compared to the baseline method, GS Only, whereas IHCCH, SS Only, Mixed and GS+SS methods performed generally worse. Nuclei detection performance on the source domain serves as our model

verification step. Our findings indicate that including SS data from the target domain does not degrade model performance on the source domain.

### 3.2. Target Domain: Nuclei Detection

We next test our method on an adaptation task as we shift from source domain to target domain pixel-level assessments. 10 TMAs from the target domain with pixel-level Ki-67$^-$ and Ki-67$^+$ expert annotations were used to test nuclei detection performance. The distribution of the F1 scores on the target domain test set is shown in Figure 5 and summarized in Table 1. The GS+SS method achieves superior performance exceeding all other methods and significantly higher performance than the baseline method regardless of the SS increment.

### 3.3. Target Domain: PI Computation

We extend the use of our approach to another adaptation task involving a change in the level of assessment, specifically from patch-level to patient-level. $\Delta$PI is assessed on 151 patients (325 TMAs) from the target domain. The distributions of the $\Delta$PI are shown in Figure 1 and summarized in Table 1. SS+GS achieves superior PI prediction performance exceeding all other methods and achieving significantly lower PI error ($p < 0.05$) compared to the baseline method, GS Only, regardless of the SS increment. The $\Delta$PI for GS only methods is $\sim 7.5\%$, but using the SS+GS method leads to a decrease in error by $\sim 3.5\%$, which is a significantly greater improvement compared to other methods. SS+GS methods also yielded the lowest $\Delta$PI standard deviation signifying less variability and more consistent and reliable predictions. As some PI intervals have greater clinical significance, the patient-level PI performance was evaluated in intervals of 10% as depicted in Figure 2. SS+GS methods maintain the lowest $\Delta$PI across all intervals (excluding 30% to 40% for UV-Net) which demonstrates optimal performance in clinically relevant ranges.

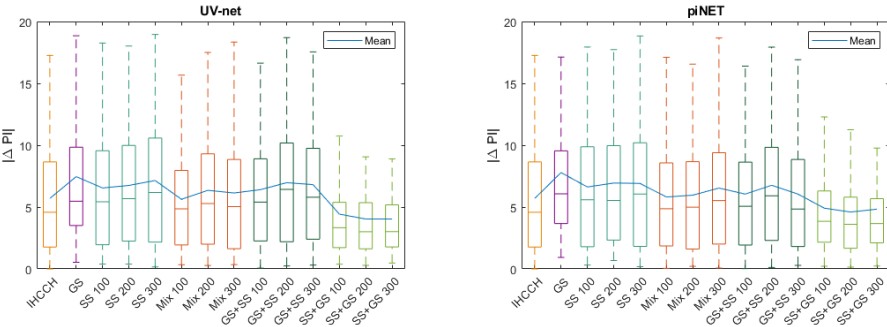

Figure 1: Patient-level $\Delta$PI for piNET and UV-Net.

### 3.4. Qualitative Evaluation: t-SNE

We analyze the effects of the models on source and target domains further with t-SNE, a popular method to visualize high-dimensional data in 2D (van der Maaten and Hinton, 2008). Figure 3 illustrates such feature visualizations from source and target images obtained from GS Only, GS+SS and SS+GS models. The features learned for the source and target

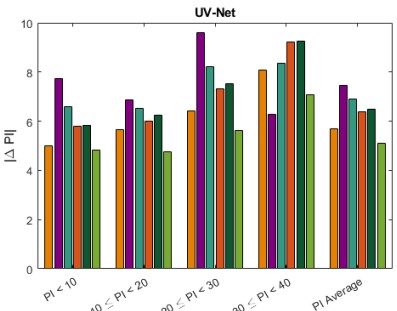
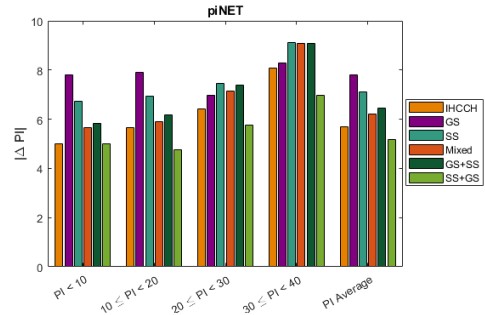

Figure 2: Mean ΔPI across 152 patients. The interval [0 10) contains 72 patients, [10 20) contains 52 patients, [20 30) contains 19 patients, [30 40) contains 8 patients.

domains in the GS only and GS+SS models are diffuse and mostly non-overlapping, which likely causes reduced generalization. However, features from the SS+GS model are similar across source and target domains, which likely resulted in improved generalization and top performance on target domain data.

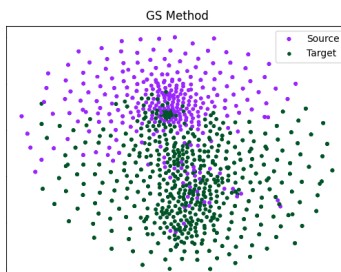
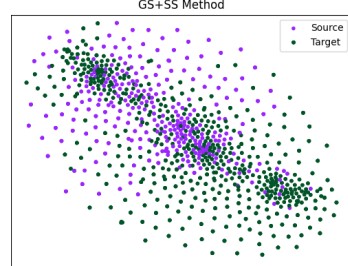
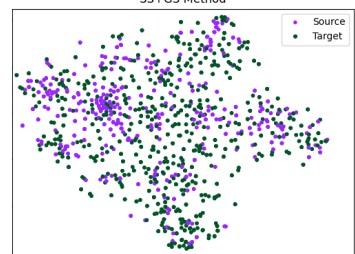

Figure 3: t-SNE plots with perplexity 15 shown from features extracted from piNET models. Features from source (purple) and target (green) are diffuse in GS Only and GS+SS methods; but similar for the proposed SS+GS method (domain gap is minimized). t-SNE hyper-parameters are consistent between visualizations.

## 4. Discussion

Ki-67 PI is visually assessed by pathologists to estimate prognosis (Petrelli et al., 2015) and decide whether adjuvant chemotherapy should be added to a patient's treatment plan (Nielsen et al., 2021). A high Ki-67 proliferation index is associated with a poor prognosis (Petrelli et al., 2015) and better eligibility for adjuvant chemotherapy (Nielsen et al., 2021). The monarchE Phase 3 (Polewski et al., 2022)—establishes > 20% Ki-67 PI as a clinically relevant threshold to stratify patients with estrogen receptor-positive early breast cancer eligible for adjuvant chemotherapy. However, various preanalytical, analytical and interpretation factors affect the scoring of Ki-67 by pathologists and lead to high inter-rater variability. Automated tools, such as deep learning can be used to bring objectivity and efficiency, thus improving the clinical utility of Ki-67 scoring.

While more accurate than other tools, deep learning methods experience a reduction in performance when applied to out-of-domain data. Covariate shifts between source and target domains are common in digital pathology due to different staining protocols and scanning equipment/software. This presents a significant challenge for clinical translation, as the current industry standard is to train models using data only available to the vendor. To address this issue and move closer to widespread deployment, this work presents an unsupervised domain adaptation method for Ki-67 quantification to focus on creating models that generalize to target data. The proposed pipeline learns the target distribution without manual annotations, which would be time-consuming and costly to obtain for medical images. Pseudo labels (SS labels) are extracted from the target domain in an unsupervised manner using the IHCCH method, and this data is used to supplement training datasets to learn domain- and problem-specific features. This framework can be easily implemented at any target site as a laboratory-specific calibration method, which can simplify deployment not only for Ki-67 quantification but also for a wide range of medical imaging applications.

We evaluated five training configurations (GS Only, SS Only, Mixed, GS+SS, SS+GS) on two Ki67 architectures (piNET and UV-Net) and found improved performance, particularly for the SS+GS configuration compared to the baseline, GS only. This suggests that although the SS labels may be slightly noisy (F1 score of 0.53 on source and 0.57 on target), incorporating data from the target domain can help the models learn domain-specific features. This was evident from the t-SNE plots, which showed a clear overlap in features learned for the target and source distributions in the SS+GS models. On the other hand, the GS+SS models did not perform as well, despite being the standard practice in the community. We believe that fine-tuning with the noisy SS labels forces the model to remember the noise more prominently. However, in the SS+GS configuration, the model was first trained with the noisy SS labels and then refined with clean GS labels, leading to better performance and an overall PI accuracy of 95.9% achieved using piNET. Furthermore, across clinically relevant PI ranges, the SS+GS models exhibited the best performance and demonstrated consistency (low standard deviation across multiple training runs).

We recognize there is ample opportunity to enhance performance and gain a deeper understanding of the impact of SS labels. Our strategy includes enhancing pseudo label generation, refining patch selection, diversifying patient cohorts, and assessing SS label source domain effects. We'll also compare our approach to domain adversarial learning and self-supervised model distillation. Future studies will explore per-site calibration in other datasets and benchmark against state-of-the-art methods.

## 5. Conclusion

In this study, we address the problem of domain adaptation for automated Ki-67 quantification in invasive breast cancer. We present a novel self-supervised approach that shows that using target domain pseudo labels (SS) for pre-training and fine-tuning with ground truth (GS) data from the source domain leads to improved performance on both source and target domains. The proposed method enhances the robustness of AI models to domain variations and improves adaptation to unseen data distributions. The training pipeline overcomes the difficulties of scarce labelled data and costly manual annotations; a challenge in medical imaging applications. These findings can drive widespread clinical utilization of automated quantification tools in digital pathology.

## Acknowledgments

We acknowledge the Canadian Cancer Society, and MITACs Canada for funding this research.

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

## 6. Appendix

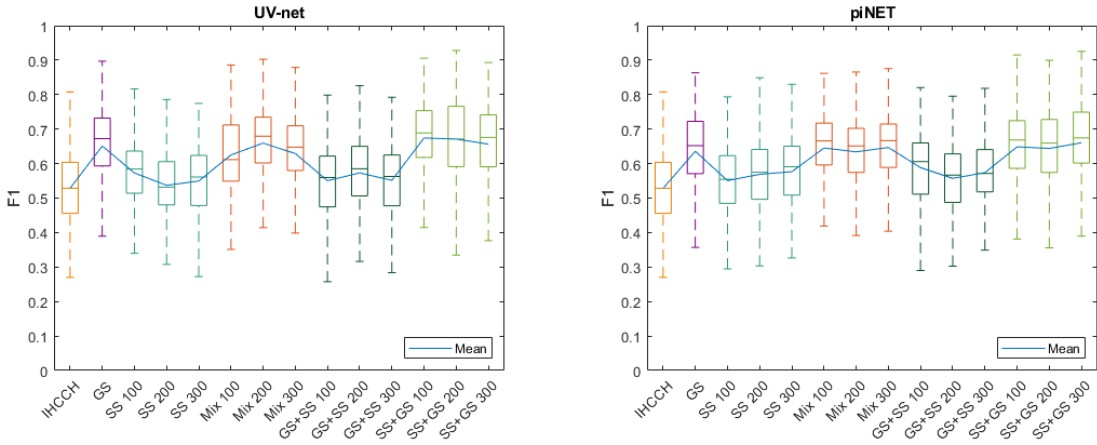

Figure 4: F1 scores for piNET and UV-Net on source dataset.

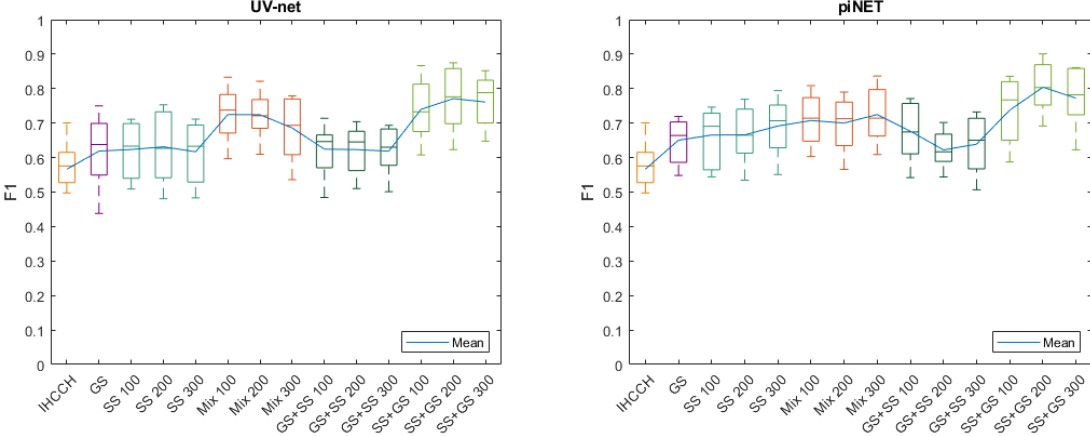

Figure 5: F1 scores for piNET and UV-Net on the target dataset.

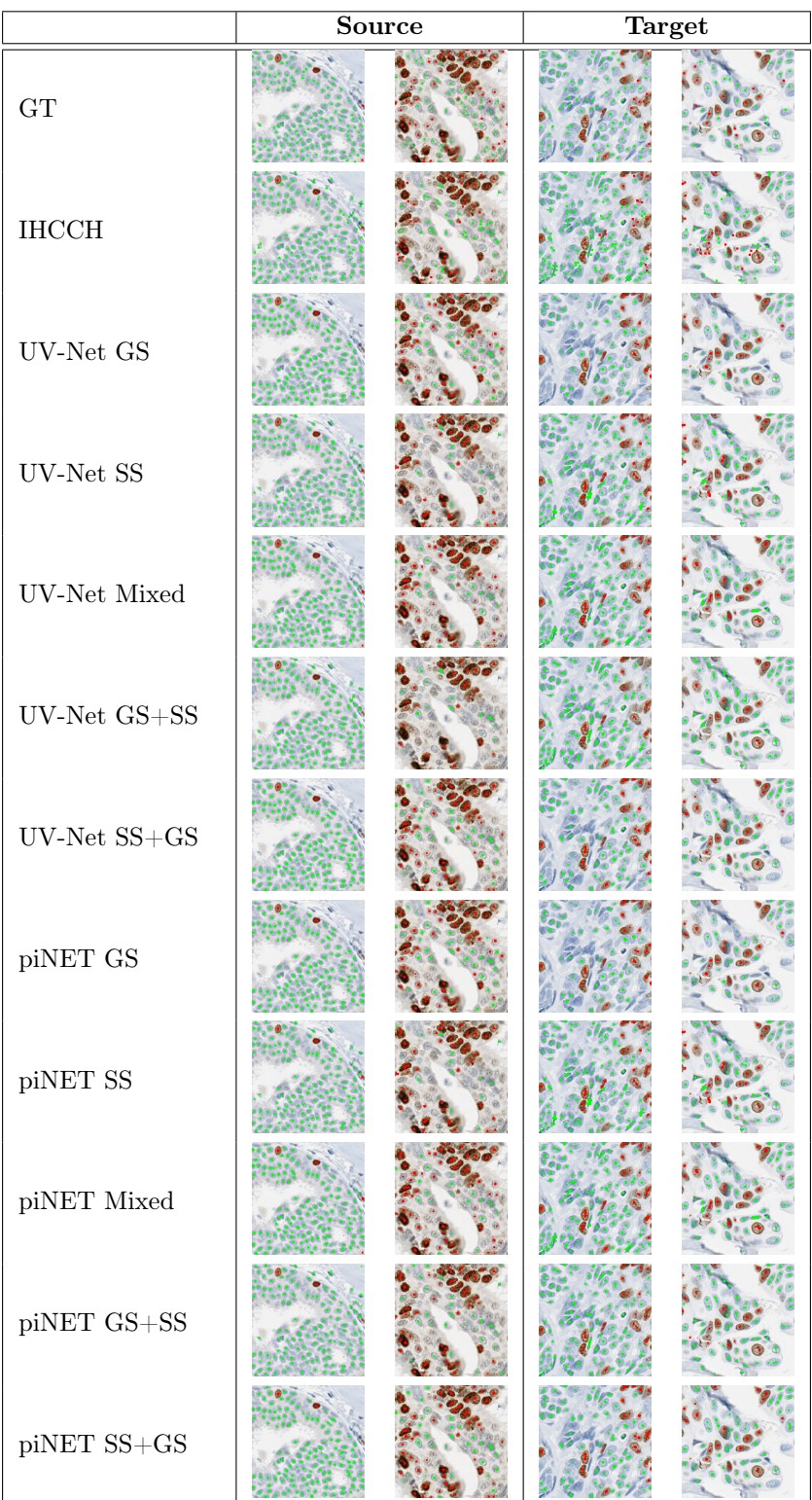

Figure 6: Visualization of model predictions on source and target domains. Green: Ki-67$^-$, Red: Ki-67$^+$.

Table 2: Summary of the patches used for model training, tuning, and testing. GS labels are from the source domain (SMH and Deepslides). SS labels are from the target domain (UHN).

| Model | SS Train | GS Train | SS Tune | GS Tune | Source Test | Target Test |
|-------|----------|----------|---------|---------|-------------|-------------|
| IHCCH | 0 | 0 | 0 | 0 | 170 | 20800 |
| GS | 0 | 340 | 0 | 0 | 170 | 20800 |
| SS 100 | 100 | 0 | 0 | 0 | 170 | 20800 |
| SS 200 | 200 | 0 | 0 | 0 | 170 | 20800 |
| SS 300 | 300 | 0 | 0 | 0 | 170 | 20800 |
| Mixed100 | 100 | 340 | 0 | 0 | 170 | 20800 |
| Mixed200 | 200 | 340 | 0 | 0 | 170 | 20800 |
| Mixed300 | 300 | 340 | 0 | 0 | 170 | 20800 |
| GS+SS 100 | 0 | 340 | 100 | 0 | 170 | 20800 |
| GS+SS 200 | 0 | 340 | 200 | 0 | 170 | 20800 |
| GS+SS 300 | 0 | 340 | 300 | 0 | 170 | 20800 |
| SS+GS 100 | 100 | 0 | 0 | 340 | 170 | 20800 |
| SS+GS 200 | 200 | 0 | 0 | 340 | 170 | 20800 |
| SS+GS 300 | 300 | 0 | 0 | 340 | 170 | 20800 |

Table 3: The average standard deviation was calculated across 3-fold cross-validation models. The standard deviation is a measure of the robustness and reproducibility of the training configurations when using different folds of the dataset and varying randomization parameters. The F1 target scores and $\Delta$ PI remain relatively stable across all models.

| Model | UV-Net | | piNET | |
|-------|--------|--------|--------|--------|
| | F1 Target | $\Delta$PI | F1 Target | $\Delta$PI |
| GS | 4.7e-2 | 1.6 | 4.0e-2 | 2.9 |
| SS | 4.9e-2 | 1.1 | 4.4e-2 | 1.0 |
| Mix | 5.3e-2 | 1.1 | 5.9e-2 | 1.0 |
| GS+SS | 3.5e-2 | 1.2 | 5.2e-2 | 1.2 |
| SS+GS | 4.9e-2 | 1.3 | 4.9e-2 | 1.1 |

