# OpenReview forum: "Domain Adaptation using Silver Standard Labels for Ki-67 Scoring in Digital Pathology A Step Closer to Widescale Deployment"
_MIDL.io/2023/Conference — MIDL 2023 Poster_

### Official Review · Reviewer_PNgq · 2023-02-01

**Confidence:** 4
**Preliminary Rating:** 2
**Recommendation:** Poster

**Summary:**

This paper presents a framework for unsupervised domain adaptation of deep neural networks for histopathology image analysis. The framework is presented as a more general approach to training of models in histopathology and then demonstrated for the problem of Ki-67 IHC quantification in breast cancer histopathology. The authors hypothesise that pertaining on silver standard annotations allows the model to learn low-level features that offer improved performance on the target domain.

**Strengths:**

The main idea behind the unsupervised domain adaptation is generally well presented (but can be further improved, see my comments below). The approach itself is relatively straightforward and well-motivated. The experimental setup is well executed.

**Weaknesses:**

The main weakness is that the authors did not investigate if their hypothesis (that SS allows the network to learn low-level features that generalise better) is true. While the results show improved performance of the "tuning approach" it is not clear if this was due to learning such low-level features. It is also unclear if the performance improvements are clinically significant for this task.

**Deanonymize Review:**

no

**Detailed Comments:**

While the text is very easy to follow, I find that the organization/placement of the figures makes it very difficult to evaluate the results that they present. I would encourage the authors to add a table with the main results in the main text (and not in the appendix).

**Paper Type:**

methodological development

**Questions To Address In The Rebuttal:**

- Evaluate the main hypothesis.
- Comment on the clinical significance of the improvement.
- It would be interesting to see if the histogram-based method for setting the SS evaluated as an additional baseline (maybe it works well by itself?).

---

### Official Review · Reviewer_BGtw · 2023-02-04

**Confidence:** 4
**Preliminary Rating:** 4
**Recommendation:** Poster

**Summary:**

The authors present experiments on deep learning-based Ki-67 quantification for breast cancer in the context of transfer learning. The approach considers two annotation qualities: expert annotated (gold standard - GS) from the source domain and fully-automatic image/color-analysis approach (silver standard - SS) from the target domain. Different training and transfer learning strategies utilizing these datasets are evaluated with the goal to improve performance on the target domain. The authors report equal to better performance for F1 score w.r.t. cell detection and improved performance for mean Ki-67 proliferation index error when they train initially on the SS data and fine-tune on the GS data compared to training on GS, SS or mixed data. No model distillation or other domain adaptation approaches are used.

**Strengths:**

- the paper is well structured and clearly written, the presented approaches are straight forward
- evaluation on two different network architectures
- within the experimental setting that the authors have chosen, they report diligently on different variants of mixing data from the "source" domain and the "target" domain
- the evaluation, the results and the experiments are well visualized and analyzed (including information in the appendix)
- the performance of the algorithm is evaluated both on a per-cell level (F1-score) and on the patient level ($|\Delta PI|$)

**Weaknesses:**

Disclaimer: I have not considered/looked into the Appendix (13 pages) in detail if some the information is potentially provided there.

- the paper does not present a considerable methodological novelty - which by itself is not a problem for a validation paper. However, from my perspective, the _training_ dataset is still fairly small for a validation study (2 (GS) +1 (SS) domains, <1k patches sized 256²) and - for me more critical - the authors leave out obvious experiments (see detailed comments)
- the quality of the generated "silver standard" is not evaluated in the main paper (this could have been done also on the 10 patches that were expert-annotated)
- given the fairly small dataset and the variable performance for the different samples reported on the GS dataset, I am generally wondering regarding the robustness, quality and the reproducibility of the training; however, no repeated trainings are reported.
- I am missing a discussion on how clinically impactful the mean error reported in Fig. 2 is in terms of inter-/intra-rater variability and how the currently reported error would change the assessment of these images

**Deanonymize Review:**

no

**Detailed Comments:**

As mentioned above, I have not looked into the Appendix (13 pages) in detail, unless this was explicitly linked in the main document, in which cased I briefly looked at this information. With a manuscript with a total of 24 pages, a detailed review including the information there is not feasible with the review period given for this paper. I also interpret the author & reviewer guidelines by MIDL such that this is also not the purpose of the supplementary material.
I feel that some of the content in the appendix is fairly important to make the paper self-contained but is listed only there (e.g., visualization of the pipeline, F1-performance). There at least the main quantitative insights should also be mentioned in the main text (e.g., missing for 3.1).

The authors call their approach "Unsupervised Domain Adaptation", which I believe is not a good choice here:
- the authors do not adapt to a specific domain, but rather use a form of pretraining using weak(er) labels (silver standard) in the target domain, which is then fine-tuned the source domain
- the automatic labeling approach generating weak(er) or pseudolabels does not fall into the typical UDA setting, where no labels are available (e.g., domain adversarial training, metric learning, non-contrastive learning); this is not discussed.
- given the order of training, the notion of "source" and "target" domain is somewhat weakened; this deviation from the typical terminology is not further explained

Following this, I am missing a couple of experiments that I would see as necessary and/or straight forward:
- in typical transfer learning, a network trained on the source task is adapted to the target task; Why is a transfer GS->SS not included? For me, this also lends itself to a clearer adaptation across multiple datasets, i.e., having a well-trained & validated "GS"-model which can then be adapted to different target datasets
- here, an evaluation (and confirmation) of the trends on different target datasets would have also made sense for a paper centering on validation (potentially even a "fine-tuning" using SS annotation of additional data from the source domain to see the (additional) benefit of weakly annotated data
- at least from the title, I would have also expected at least one standard DA approach (e.g., domain adversarial learning) to be included in the evaluation - if the focus was on different target datasets, this could have also been fine.
- the performance on the GS dataset seems to be highly variable and not particularly good, even when trained on the GS dataset (F1-score of 0.6). This is not discussed, and the results seem to be reported on a single training of each network variant. Given that there is partially considerably difference between similar settings (e.g., 5-10 %points for Tuned 200 vs. Tuned 250), why were these experiments not repeated?

**Paper Type:**

validation/application paper

**Questions To Address In The Rebuttal:**

- What is the motivation for only including an SS->GS fine-tuning in the experiments and not a GS->SS finetuning?
- What is the clinical impact of the reported performance/error margins? To what extent is this assessment "ready" for widespread clinical assessment?
- Did the authors investigate the reproducibility of their results? How robust are the trends under multiple repetitions?

---

### Official Review · Reviewer_RoEe · 2023-02-06

**Confidence:** 4
**Preliminary Rating:** 2

**Summary:**

The paper describes a method to overcome the problem of domain shift in analysis of pathology images.  The authors consider a situation where gold standard labels for tumor cells are available on breast tissue sections from one center but no labels exist on data from a second center.  They demonstrate that "silver standard" (SS) labels can be generated on the data from center 2, by basic image processing techniques, and experiment with methods to use these SS labels in training.

**Strengths:**

The paper is written in clear English and tackles the well-known and ongoing issue of domain shift.  Experimental descriptions are adequate and the authors have performed multiple analyses of their various experiments.

**Weaknesses:**

One main weakness is that the majority of the figures and all of the tables are contained in the appendices of the paper, and it does not read as a self-contained 8 page paper.  It would be impossible to understand or evaluate the paper without continuous reference to the appendices and therefore I feel it has not been structured correctly for MIDL.   Tables and figure in the appendices are referred to from the beginning of the paper, while those 2 results figures actually included in the paper are only referenced on pages 6 and 7.

The figures in the appendices are too many and too similar to each other - the authors need to choose the most relevant and important information that they want to present and put it in the main body of the paper.

There was no substantial novelty that I could find in the paper.  From the introduction I understood that the authors proposed pre-training with silver-standard labels as a new approach, compared to re-training with silver-standard labels, as done by other authors.  This is a small novelty, but in the experimental set-up the approach of others (re-training with silver-standard) was not shown, so no direct comparison was performed with the described method from existing literature.

The approach the authors describe means that the entire network has to be retrained from scratch for each new domain (which is obviously complicated, if for example the original training data is not available or has privacy limitations), while the "re-training" method from literature does not suffer from this drawback.  Therefore it is essential that the authors demonstrate the advantage of their approach over that one.

The nuclei detection experiments did not show any advantage to pre-training with silver-standard(SS) labels (and then tuning with gold-standard(GS) labels) or just training from scratch with a mix of SS and GS labels.   The novelty of the work is therefore very limited.

**Deanonymize Review:**

no

**Detailed Comments:**


The text in the paper could be shortened and simplified in many cases to make room for simple data tables and pipeline diagrams.  For example, section 2.3 provides a reasonably lengthy description of an image processing method that is apparently already detailed in another work.  The authors could provide a short sentence indicating that basic image processing methods such as thresholding are used, refer to the other paper, and include the diagram which illustrates the method nicely.

In the description of the data I could not find any information about how many positive and negative centroids were included in the various datasets.

Section 3.4 (post-hoc analysis) does not add a lot to the paper in my view and could have been excluded as part of an effort to tighten up the content and make a more concise and focused 8 page paper.



**Paper Type:**

both

**Questions To Address In The Rebuttal:**

The major difficulty I have with the paper is the structure - the authors need to limit the experiments and different types of plots and focus on conveying their entire message within 8 pages.  The appendix should contain only additional information for the interested reader.

Secondary to that is the limited novelty of the work and the fact that it is not compared with the method of training on GS and fine-tuning on SS.

---

### Meta-Review · Area_Chair_Tpd1 · 2023-02-23

**Recommendation:** Reject
**Confidence:** 5

**Metareview:**

Although the authors have tried to improve the paper's writing and presentation. I applaud their efforts and their interaction with the reviewers during the rebuttal period. After reading the final revised version I also still find some sections very difficult to understand (mostly the Appendix having most of the information). Using pseudo labels to train a network and optimize further with fewer gold standard labels has been investigated in the literature. Self-supervised, or semi-supervised methods, try to address this problem. The baseline architectures investigated are also not novel as they have been proposed previously. Finally, the method that is used for SS label generation has a very low accuracy which raises the question of how reasonable is it to use such an approach. Overall, interesting work with lots of room for improvement. I hope the authors take all the feedback into account and further improve their work. In its current form, it is not suitable for presentation at the MIDL conference.  I would welcome PCs input as I am between borderline accept/reject.

---

### Meta-Review · Program_Chairs · 2023-03-01

**Recommendation:** Accept (Poster)
**Confidence:** 5

**Metareview:**

In the discussion of the PC it was felt that the paper has enough novelty in the clinical application setting, so given the overall borderline scores and reviews, the PC decided to accept this as a Poster.